# The Effects of the Association Between a High-Fat Diet and Physical Exercise on BDNF Expression in the Hippocampus: A Comprehensive Review

**DOI:** 10.3390/life15060945

**Published:** 2025-06-12

**Authors:** Francisca Tayná da Silva Gomes, Antônio Vicente Dias de Andrade, Paloma Katlheen Moura Melo, Roque Ribeiro da Silva Júnior, Débora Lopes Silva de Souza, Élyssa Adriolly Freitas Tavares, Ingrid Garcia de Sena, Thales Allyrio Araújo de Medeiros Fernandes, Paulo Leonardo Araújo de Góis Morais, Ivana Alice Teixeira Fonseca, Cibele dos Santos Borges, José Rodolfo Lopes de Paiva Cavalcanti

**Affiliations:** 1Faculty of Health Sciences, Department of Biomedical Sciences, Universidade do Estado do Rio Grande do Norte, Mossoró 59633-010, Brazil; antoniovicente@uern.br (A.V.D.d.A.); palomakatlheen@hotmail.com (P.K.M.M.); roquejunior@alu.uern.br (R.R.d.S.J.); deboraalopes7@gmail.com (D.L.S.d.S.); elyssa20231001735@alu.uern.br (É.A.F.T.); ingridsena@alu.uern.br (I.G.d.S.); thalesallyrio@uern.br (T.A.A.d.M.F.); pauloleonardo87@hotmail.com (P.L.A.d.G.M.); ivanateixeiraf@gmail.com (I.A.T.F.); rodolfolopes@uern.br (J.R.L.d.P.C.); 2Center for Biological and Health Sciences, Department of Biosciences, Universidade Federal Rural do Semi-Árido, Mossoró 59625-900, Brazil; cibele.borges@ufersa.edu.br

**Keywords:** BDNF, high-fat diet, hippocampus, physical exercise

## Abstract

High-fat diets, characterised by their high lipid content, have been associated with structural and functional alterations in the hippocampus of rodents, including a reduction in the expression of brain-derived neurotrophic factor (BDNF), which negatively impacts learning and memory. Evidence suggests that these impairments may be attenuated by regular physical exercise. This review aimed to gather relevant scientific evidence available on the effects of the association between high-fat diets and physical exercise on BDNF expression levels in the hippocampus of rats. The studies analysed highlight a neuroprotective effect of exercise, capable of positively modulating BDNF levels and, consequently, improving the cognitive functions of these animals.

## 1. Introduction

High-fat diets (HFDs) are characterised by a high lipid content, accounting for more than 30% of total energy intake, in contrast to standard diets, which typically contain higher proportions of proteins and carbohydrates [1,2,3]. Several studies have demonstrated the negative effects of the prolonged consumption of such diets on various tissues and organs, including the liver, adipose tissue, skeletal muscle, pancreas, and heart [4,5]. In the brain—particularly in the hippocampus—HFD intake has been associated with reduced levels of brain-derived neurotrophic factor (BDNF), leading to structural and functional changes in this region [3].

For decades, HFDs have been widely used as an experimental model for inducing obesity in rodents [6]. This model is recognised for faithfully mimicking the metabolic and physiological alterations observed in human obesity, which is frequently linked to environmental factors such as physical inactivity and the excessive intake of ultra-processed foods [7,8].

Excessive HFD consumption, particularly diets rich in saturated fats, has been identified as one of the main risk factors for the development of Non-Communicable Chronic Diseases (NCDs), due to its deleterious effects on metabolism and systemic physiology [7,8].

Regular physical exercise, in turn, represents an important non-pharmacological strategy for preventing and managing the adverse metabolic effects associated with HFDs. In addition to its systemic benefits, physical exercise promotes neuroprotective adaptations, such as increased neurogenesis, enhanced synaptic plasticity, and elevated BDNF levels in the hippocampus, which directly contribute to the preservation and improvement of cognitive function [9,10,11].

Studies conducted in animal models provide evidence that HFD consumption can lead to cognitive deficits, which may be attenuated by regular physical exercise [12,13,14]. In this context, the present study aimed to discuss the effects of the combination of a high-fat diet and physical exercise on the modulation of BDNF expression in the hippocampus of rats.

## 2. Materials and Methods

To ensure a comprehensive and accurate search, the following databases were used: PubMed, Web of Science, ScienceDirect, Scopus, and Ovid Medline. The search strategy was developed based on controlled vocabularies—Medical Subject Headings (MeSH) and Emtree—combined using the Boolean operators AND and OR, as detailed in Table 1.

A total of 4836 journals that corresponded to the descriptors mentioned above were identified in the aforementioned databases, from January to May 2025. This survey allowed for a broad collection of relevant information, ensuring that the data set was representative and consistent with the criteria previously defined for the systematic search.

To ensure the relevance and methodological consistency of the studies included in this review, specific inclusion and exclusion criteria were established. Experimental studies using animal models—particularly rodents (rats and mice)—that investigated the association between high-fat diets, physical exercise, and BDNF expression in the hippocampus were included.

Excluded works comprised editor letters, preprints, and undergraduate theses (grey literature).

Journals published between 2005 and 2025 were considered eligible. No restrictions were applied regarding the language of the articles, since this criterion was not used as a filter for selection. This approach allowed for the inclusion of a greater diversity of publications, covering studies from different linguistic contexts and contributing to the expansion of the evidence base analysed. In this way, we sought to ensure that this review included a broad and global perspective, reflecting the plurality of scientific knowledge available on the topic in question.

The study selection process was conducted in three phases. Initially, two independent reviewers screened the titles and abstracts. In the second phase, the eligibility criteria were applied to the previously selected studies, and in the third phase, the articles that met the criteria were read in full.

## 3. Results

### 3.1. Hippocampus

In humans, the hippocampus is a structure located in the medial temporal lobes, forming part of the limbic system and playing a fundamental role in memory consolidation. Its curved shape, which resembles a seahorse, explains the origin of its name: from the Greek hippo (horse) and kampos (sea monster) [15,16].

In rodents, the hippocampus appears as an elongated, curved “C-shaped” structure located at the dorsomedial end, near the septal nuclei, curving beneath the diencephalon in the region of the caudal temporal cortex [17,18].

Anatomically, the hippocampus is divided into three parts: the head, body, and tail. It also presents two distinct functional subregions: the dorsal portion, which is associated with spatial memory [19], and the ventral portion, which is involved in emotional responses and stress modulation mechanisms [20]. These divisions observed in rodents correspond to the posterior (dorsal) and anterior (ventral) subregions in the human brain [21].

Functionally, the rodent hippocampus can be segmented along the dorsal and ventral axes, commonly referred to as the dorsal hippocampus (DH) and ventral hippocampus (VH). Its main regions include the dentate gyrus (DG) and the three subfields of the Cornu Ammonis: CA1, CA2, and CA3 [22].

The DG represents the primary gateway for incoming information, receiving afferents mainly from the entorhinal cortex. Its efferents project to the CA3 and CA2 regions, which in turn connect to CA1, completing the trisynaptic pathway of the hippocampus [22,23,24].

In addition to its essential role in memory consolidation, the hippocampus is also involved in the modulation of emotions, including anxiety and stress [20,25]. Evidence from various studies suggests that the broad role of the hippocampus in both cognitive and emotional domains is related to its central location in the cerebral cortex and its internal anatomical organisation [26].

It is well established that learning and memory processes are fundamental to promoting neuronal plasticity, thereby contributing to improved cognitive capacity [27]. In particular, the functional plasticity of the hippocampus stands out, as it is closely associated with synapse formation and is considered essential for memory consolidation and learning [28].

Several studies involving physical exercise demonstrate a direct relationship between this practice and cognitive benefits [1,2,3]. In a study involving 40 male C57BL/6 mice, 4 weeks of age, the animals were randomly divided into two groups: normal diet (CON, *n* = 10) and high-fat diet (HFD, *n* = 30). The HFD contained 60% fat. The animals were induced to obesity for a period of 8 weeks and subsequently randomly redistributed into three groups: HFD (*n* = 10), HFD with low-intensity training (HFDLT, *n* = 10), and HFD with high-intensity training (HFDHT, *n* = 10). The mice in the groups subjected to physical training performed treadmill exercises 5 times a week for 8 weeks, maintaining the regular consumption of their respective diets. After the training period, an assessment of cognitive function was performed using the Y-maze test, which aims to analyse the animal’s spatial memory based on the supervised exploration of the three arms of the maze. The results of the study demonstrated that the physical exercise intervention significantly improved the cognitive function of the animals that consumed a high-fat diet. The number of entries and alternations was significantly higher in the physical training groups compared to the sedentary HFD group (*p* < 0.05) [12].

In a similar study conducted by Kim et al., 40 male C57BL/6 mice, 4 weeks of age, were also used and randomly divided into four groups: a control group (CON), control group with exercise (CON + EX), high-fat diet group (HFD), and high-fat diet and exercise group (HFD + EX). Subsequently, the Y-maze test was performed, and it was observed that the percentage of spontaneous alternation in the animals was lower in the group submitted to the high-fat diet compared to the control group [29].

These findings demonstrate that the hippocampus may be impacted by the consumption of high-fat diets, significantly compromising the animals’ spatial memory. In the Y test, it was possible to naturally analyse the rodent’s exploratory capacity, observing the order of the input sequence and identifying the number of hits or misses during the test. With the consumption of an HFD, BDNF levels tend to decrease, which results in the impairment of cognitive capacity related to learning and memory. On the other hand, physical training has proven to be increasingly efficient in minimising the damage caused by this type of diet.

### 3.2. Brain-Derived Neurotrophic Factor (BDNF)

Brain-derived neurotrophic factor (BDNF) is a neurotrophin belonging to a family of neurotrophic proteins, which include nerve growth factor (NGF), neurotrophin-3 (NT3), and neurotrophin-4 (NT4) [30]. BDNF plays an essential role in neuroplasticity, particularly in regions such as the hippocampus [26]. BDNF can be found in higher concentrations in specific areas of the brain (Figure 1).

BDNF synthesis begins in the endoplasmic reticulum in the form of pre-pro-BDNF. This precursor molecule is converted into pro-BDNF with a molecular weight ~32 kDa following the removal of its signal peptide (Figure 2). Glycosylation at residue N123 within the prodomain facilitates its cleavage, which may occur either intracellularly—through the action of furin and convertase 7 in the Golgi apparatus—or extracellularly, mediated by tissue plasminogen activator (tPA), plasmin, and matrix metalloproteinases. Mature BDNF (~13 kDa) is then secreted via interaction with the sortilin receptor [28,31].

In neurons, BDNF is produced in the cell body and transported in vesicles to the axon terminals for release into the synaptic cleft. In this region, it binds to the receptors located on the dendrites of postsynaptic neurons. Additionally, BDNF can also be released through retrograde signalling, where it is secreted from dendrites to presynaptic receptors [32].

BDNF binds to two types of receptors: tropomyosin receptor kinase B (TrkB) and the p75 neurotrophin receptor (p75NTR). Mature BDNF has a greater affinity for TrkB, promoting neuroplasticity and cell survival. Pro-BDNF, on the other hand, preferentially interacts with p75NTR, which is associated with cellular apoptosis [33,34].

Cai et al. [14] demonstrated that increased BDNF expression favours structural and functional changes in neuronal networks, which are essential for synaptic plasticity. Ganguly and Poo [35] reinforce that BDNF, in addition to promoting neuronal survival, strengthens existing connections and stimulates the formation of new synapses.

It is important to note that experimental studies have observed varying conditions regarding BDNF and its relationship with neuronal plasticity. The study conducted by Gao et al. [36] aimed to evaluate the effects of electroacupuncture on the expression of BDNF, mammalian target of rapamycin complex 1 (mTORC1), and proteins associated with dendritic spine synapses in the prefrontal cortex of rodents, with the goal of regulating synaptic plasticity and alleviating depressive symptoms.

Thirty-six male Sprague-Dawley rats were randomly assigned to four groups: control, model, electroacupuncture (EA), and scopolamine (SCOP). Depression was induced through daily exposure to mild and unpredictable stressors. The EA group received electroacupuncture sessions for 20 min per day over 14 days, while the SCOP group received intraperitoneal injections of 25 µg/kg scopolamine every 16 h for 14 days.

The results showed that, compared to the control group, the model group exhibited a significant reduction in sucrose preference, increased latency to feed, and a decreased expression of BDNF, mTORC1, p-mTORC1, PSD95, synapsin I, and GluR1, along with reductions in total, mature, and immature dendritic spine densities. Conversely, EA or SCOP treatment increased sucrose preference and reduced feeding latency, as well as elevating the expression of BDNF, mTORC1, p-mTORC1, PSD95, synapsin I, and GluR1 in the prefrontal cortex. Furthermore, both treatments increased total and immature dendritic spine density, with the EA group showing a significant increase in filopodia-like spines and the SCOP group in mature spines. No significant differences were observed between the EA and SCOP groups. These findings suggest that electroacupuncture alleviates depressive symptoms in rats subjected to the CUMS model, likely through the activation of the BDNF/mTORC1 pathway and the upregulation of synaptic proteins, thereby promoting synaptic plasticity in the prefrontal cortex.

Following a similar line of investigation, Zhang et al. [37] conducted a study to evaluate the antidepressant effects of timosaponin B-III (TB-III) and its mechanisms of action. Female mice were used in a postpartum depression (PPD) model induced by the administration of sodium dexamethasone phosphate during the gestational period. Animals were allocated into control, model, fluoxetine, and TB-III groups receiving different doses (10, 20, or 40 mg/kg).

Depressive behaviours were assessed, along with serum and hippocampal cytokines such as tumour necrosis factor (TNF)-α and interleukins (IL)-1β, IL-6, and IL-10. Additionally, the levels of proteins associated with synaptic plasticity in the hippocampus were analysed, including those of BDNF, GSK-3β, GluR1, PSD95, and synapsin I. The results showed a significant increase in TNF-α, IL-1β, IL-6, and IL-10 in the model group, along with a reduction in serum IL-10 and the hippocampal expression of BDNF, GSK-3β, GluR1, PSD95, and synapsin I compared to the control group. Treatment with fluoxetine or TB-III significantly reversed these changes in cytokine and protein levels. Thus, timosaponin B-III demonstrated protective effects against postpartum depression, possibly mediated by the modulation of inflammatory cytokines, the activation of the BDNF pathway, and the regulation of proteins associated with synaptic plasticity.

In this same context, experimental studies indicate that reduced BDNF levels are associated with diminished neural adaptability to new stimuli, resulting in impairments in memory and learning [2,38]. This deficiency is observed in various neurodegenerative and psychiatric conditions.

On the other hand, regular physical exercise has consistently been associated with increased BDNF levels. Feng et al. [39] investigated the effects of aerobic exercise on dendritic structure in the hippocampus and cerebral cortex, as well as on the BDNF-mTOR signalling pathway in ovariectomised (OVX) rodents. The animals were divided into four groups of ten: control (sham), control with exercise, OVX, and OVX with aerobic exercise. The results demonstrated that aerobic exercise significantly improved dendritic morphological changes, increased the expression of proteins related to synaptic plasticity (PSD95 and GluR1), and activated the BDNF-mTOR signalling pathway in both the hippocampus and cerebral cortex. Exercise also reversed the dendritic alterations induced by ovariectomy, indicating that its beneficial effects may be mediated by the activation of the BDNF-mTOR pathway.

In a similar study, Liu et al. [40] conducted an experiment to evaluate the effects of resistance training on adaptive mechanisms in the cerebral cortex of aged rodents. Male Sprague-Dawley (SD) rats, free from pathogens, of different ages (3 months, *n* = 20; 13 months, *n* = 24; and 23 months, *n* = 24) were assigned to a sedentary or resistance training group for 10 weeks. The intervention involved treadmill running with progressive intensity, ranging from 60–65% to 70–75% of the maximum oxygen consumption.

Analyses revealed age-related structural changes in the cerebral cortex. Superoxide dismutase (SOD) expression declined with age, while the expression of BDNF, SYN1, and CaMKIIα increased. There was also a slight increase in the mRNA levels of AMPKα1, SirT2, IP3R, AKT1, and mTOR. Resistance training enhanced the expression of SOD, BDNF, and synaptophysin (SYN1) in the cerebral cortex, with more pronounced effects in the older animals. The intervention significantly increased BDNF expression, positively regulated CaMKIIα signalling, activated the AMPK and IP3R/AKT1/mTOR pathways, and consequently improved synaptic plasticity in the cerebral cortex.

Therefore, physical exercise plays a central role in both central and peripheral mechanisms. Huang et al. [41] demonstrated that acute and chronic exercise sessions activate molecular cascades that result in increased BDNF expression, promoting improvements in cognition. This phenomenon is partially mediated by enhanced cerebral blood flow, which supports tissue oxygenation and the activation of molecular mechanisms associated with synaptic plasticity [42]. These findings reinforce the importance of BDNF as a key mediator of neuroplasticity and a relevant marker for investigating structural and functional adaptations in the nervous system.

### 3.3. Pathophysiological Effects of High-Fat Diets

The lipids consumed in the diet are most commonly in the form of triglycerides (TGs), composed of a polar end—a glycerol molecule—and a non-polar end with three fatty acids. This composition defines them as triacylglycerols. The carboxyl and methyl functional groups attached to the carbon chain may vary in the number of unsaturations, i.e., double bonds, classifying them as saturated, monounsaturated, or polyunsaturated, and may also differ in chain length. These variations in length and the degree of unsaturation give fatty acids different physical, chemical, and biological properties in terms of their utilisation by the body [43].

It is important to note that the effects of HFDs may vary depending on the lipid composition of the diet. The ratio of saturated, unsaturated, omega-3, and omega-6 fatty acids directly influences metabolic and neurological outcomes [44,45]. This highlights the importance of considering not only the total fat content but also its nutritional quality.

In rodent studies, fat content in experimental diets generally ranges from 45% to 60%, depending on the objectives and the metabolic model being induced [1,2,3]. Several studies show the high-fat diet with a percentage of 60% fat as one of the most used, due to its high concentration in relation to the amounts of carbohydrates and proteins, which ends up favouring an increase in the adiposity of rodents, leading to the development of obesity, dyslipidaemia, insulin resistance, and cognitive problems. These investigations seek to show the damage caused by the consumption of diets rich in fats in the metabolism of rodents [1,12,29,38,46]. In humans, a diet is considered high in fat when more than 35% to 40% of total caloric intake comes from lipids [47].

The chronic consumption of high-fat diets, especially in the absence of regular physical activity, may trigger significant pathophysiological changes in the central nervous system, with direct impacts on the hippocampus. Animal model studies show that prolonged exposure to HFDs is associated with structural and functional impairments in this brain region, resulting in relevant cognitive deficits [48,49,50].

Morphologically, HFDs have been associated with hippocampal volume reduction, decreased synaptic density, and a marked decline in proteins essential to neuroplasticity, such as BDNF [3].

Furthermore, adult neurogenesis was found to be reduced, compromising the formation and maintenance of efficient synapses and negatively affecting cognition [1,51].

Mechanisms responsible for HFD-induced reductions in BDNF include increased oxidative stress, neurogenic inflammation, and cerebral insulin resistance. Pistell et al. [52] suggest that the release of pro-inflammatory cytokines triggered by excessive fat consumption interferes with BDNF synthesis and secretion, causing structural and functional damage to brain tissue. Simultaneously, insulin resistance impairs intracellular BDNF signalling, exacerbating cognitive deficits [53,54].

The inflammatory process caused by HFDs is strongly related to the action of adiponectin and leptin, hormones that act directly on eating behaviour and neurofunctional changes in the brain. The release of adiponectin is associated with the degree of adiposity in the body, as it exerts anti-inflammatory effects on insulin resistance and dyslipidaemia. This anti-inflammatory process includes the inhibition of the inflammatory cytokines IL-6 and TNF-α. Leptin, in turn, crosses the blood–brain barrier through specific transport mechanisms and acts directly on the hypothalamic centres, regulating eating behaviour [52,55].

BDNF can protect neurons against oxidative damage; however, the consumption of HFDs can cause a reduction in BDNF levels and BDNF mRNA gene expression, as well as other proteins that act in synaptic plasticity such as synapsin I and CREB. Oxidative stress causes a reduction in BDNF levels in the brain due to lipid peroxidation, a process of lipid degradation carried out by free radicals, causing damage to neurons and negatively impacting synaptic plasticity [56].

In addition to molecular alterations, animal behaviour is also affected. Rodents fed HFDs show impairments in spatial memory, associative learning, and object recognition tasks, as demonstrated in standard experimental protocols such as the Morris water maze and object recognition tests [48,49]. Moreover, increased anxiety- and depression-like behaviours have been reported, likely related to chronic inflammation and reduced synaptic plasticity [50].

These findings underscore the relevance of animal models in understanding the neurological and metabolic impacts of chronic fat consumption, offering a basis for developing preventive and therapeutic strategies to mitigate the cognitive and behavioural damage associated with poor dietary habits [54,57].

### 3.4. Neuroprotective Effects of Physical Exercise

Animal models, such as rodents, have been widely used to investigate the mechanisms through which physical exercise exerts beneficial effects on brain function. Experimental evidence demonstrates that various exercise protocols, differing in intensity and duration, lead to significant improvements in neuroplasticity, learning, and memory [58,59].

It is worth highlighting the review by Tari et al. [60], which aimed to evaluate the neuroprotective mechanisms of resistance exercise and emphasised the importance of cardiorespiratory fitness (CRF) in promoting healthy brain ageing. Among the key findings, CRF was shown to mediate the neuroprotective effects of exercise through physiological mechanisms such as increased cerebral blood flow, reduced inflammation, and enhanced neuroplasticity.

In the same line, the review by Mahalakshmi et al. [61] analysed the effects of physical exercise on neurodegenerative diseases. It became evident that exercise confers substantial health benefits, particularly in individuals who engage in regular physical activity. On a molecular level, although many signalling mechanisms remain to be fully elucidated, evidence suggests the involvement of neurotrophins such as BDNF, as well as hormones like irisin and neurotransmitters such as dopamine. These agents are directly involved in neuronal plasticity and the protection of brain function.

In the context of Parkinson’s and Alzheimer’s diseases, physical exercise has been shown to significantly increase BDNF expression, especially in the hippocampus, a region directly associated with memory and learning. Notably, BDNF plays a central role in synaptic plasticity and neuronal stress resilience, acting as a positive regulator of these processes.

Complementarily, Tuon et al. [62] investigated the effects of physical training on neurochemical and oxidative stress markers in the brains of rodents with Parkinson’s disease. Animals were divided into trained, untrained, and sham groups, then sacrificed for an analysis of tyrosine hydroxylase (TH), BDNF, α-synuclein, sarcoplasmic reticulum Ca2 + -ATPase (SERCA II), and oxidative stress markers. The untrained and sham groups showed a significant reduction in TH, BDNF, α-synuclein, and antioxidant markers, along with an increase in oxidative stress indicators. Conversely, trained animals maintained stable TH levels; exhibited elevated levels of BDNF, SERCA II, and antioxidant markers; and showed reduced lipid and protein oxidative damage.

Arida et al. [58] also demonstrated that regular physical exercise contributes to neuronal plasticity and improved cognitive performance and that the combination of adequate intensity and duration can positively influence neuronal survival.

Various exercise modalities have been employed in rodent studies, including forced protocols (e.g., treadmill running or swimming) and voluntary activities such as wheel running [63,64,65,66]. Regardless of the modality, regular exercise is associated with increased cerebral blood flow, leading to better oxygenation and nutrient distribution, which favours neurogenesis and BDNF expression [42,67].

Additionally, physical exercise increases the production of lactate, which crosses the blood–brain barrier and accumulates in the hippocampus, stimulating BDNF expression via SIRT1 pathway activation. This mechanism has been linked to improvements in learning and memory [68].

Exercise intensity also influences cognitive effects. Wu et al. [69] reported that low-intensity treadmill exercise provided greater benefits for memory and learning compared to high-intensity exercise. However, studies on intense exercise present conflicting results: some indicate cognitive impairments, while others demonstrate improvements in cognitive function [70,71,72].

In a study by Molteni et al. [3], rats fed a high-fat diet for two months exhibited spatial memory impairments and reduced BDNF levels. The introduction of voluntary exercise significantly mitigated these effects, highlighting the neuroprotective potential of exercise against diet-induced damage.

Recent research also indicates that regular physical activity attenuates the inflammatory processes and oxidative stress induced by high-fat diets. A reduction in pro-inflammatory cytokines and an increase in endogenous antioxidant production foster a brain environment conducive to maintaining and restoring cognitive functions [73,74].

These findings reinforce the role of physical exercise as a fundamental non-pharmacological tool for promoting brain health, particularly in the face of metabolic imbalances caused by poor dietary habits [12,14].

### 3.5. The Combined Effects of High-Fat Diets and Exercise on the Hippocampus

The prolonged consumption of high-fat diets without regular physical activity can significantly compromise the structure and function of the hippocampus in rodents, resulting in cognitive impairments. These negative effects include the reduced expression of proteins associated with neuroplasticity, such as BDNF, and decreased neurogenesis, directly impacting learning and memory processes [38].

Studies have shown that even in animals maintained on high-fat diets, physical exercise—either voluntary or treadmill-based—can positively modulate BDNF levels. A relevant example is the study by Molteni et al., which used 24 female rats of the Fisher 344 line that were two months old. The animals were randomly divided into four experimental groups: regular diet/sedentary (DR + S), high-fat diet/sedentary (HFD + S), regular diet with physical exercise (DR + E), and high-fat diet with physical exercise (HFD + E). The objective was to analyse the levels of BDNF mRNA and BDNF proteins in the hippocampus. The results demonstrated that the DR + E group, subjected to voluntary wheel running for two months, presented a 135% increase in BDNF mRNA levels (*p* < 0.01) compared to the DR + S group. On the other hand, the animals in the HFD + S group presented a reduction in BDNF mRNA levels of 76% (*p* < 0.05), indicating a negative effect of the high-fat diet. Notably, exposure to physical exercise was able to increase BDNF mRNA levels in animals fed a high-fat diet, raising the values from 76% to 91% (*p* < 0.05). Furthermore, voluntary exercise promoted a significant increase in BDNF protein levels; in the group with a regular diet, the values went from 100% to 185% (*p* < 0.01), while in the group with a high-fat diet, the levels rose from 61% to 141% (*p* < 0.01) [3].

In another study involving 80 male C57BL/6 mice aged four weeks, animals were randomly divided into four groups: control (CON), control with exercise (CON + EX), high-fat diet (HFD), and high-fat diet with exercise (HFD + EX). The HFD groups received feed composed of 60% fat and underwent treadmill exercise after 20 weeks of diet. Using the Morris water maze, it was observed that spatial learning was significantly reduced in the HFD group compared to the CON group. However, performance in the HFD + EX group was significantly better than that in the HFD-only group. Regarding BDNF levels in the hippocampus, an increase in expression was again observed in the HFD + EX group compared to the HFD group without exercise [1].

These findings indicate that physical exercise can improve the hippocampal BDNF levels negatively affected by HFDs and may stimulate the increased proliferation and differentiation of hippocampal cells in animal models [75]. Physical activity activates the insulin signalling pathway in the hippocampus, favouring BDNF expression and neurogenesis in this region [76].

These data suggest that physical exercise has the potential to reverse the deleterious effects of HFD on synaptic plasticity and, consequently, on cognition. Increased BDNF levels have also been observed in studies using both wheel running and treadmill exercise protocols, reinforcing the efficacy of physical activity as a neuroprotective strategy [64].

Even high-intensity exercise, traditionally associated with increased physiological stress, has shown positive results in some studies. Mice consuming diets with 60% of calories from fat and subjected to high-intensity treadmill training demonstrated improved spatial learning, memory, and reduced inflammatory markers in the hippocampus, along with significantly increased BDNF levels [38].

One of the mechanisms proposed to explain these beneficial effects involves the role of exercise in synaptic transmission and the expression of proteins related to neuronal plasticity, including BDNF. Thus, physical exercise is considered an effective non-pharmacological strategy to counteract the cognitive deficits induced by high-fat diets [77,78].

## 4. Conclusions

In general, studies have shown that high-fat diets can significantly reduce BDNF levels in the hippocampus of rodents, negatively impacting learning and memory processes. Conversely, regular physical exercise—even in the context of high-fat dietary intake—has proven effective in increasing the levels of this neurotrophin, reversing cognitive impairments and promoting synaptic plasticity.

Performing physical activity on the treadmill 5 times a week for 8 weeks was the most commonly used duration among the studies, with no relevant differences between low- and high-intensity exercises, proving that both intensities help prevent cognitive deficits, especially in the hippocampus.

In addition to strengthening the body, physical exercise acts as a neuroprotective agent, capable of preventing and mitigating the effects of poor dietary habits. Considering the growing consumption of ultra-processed and high-fat foods in the human population, strategies to promote physical activity are increasingly relevant to public health, especially for the prevention of neurodegenerative diseases.

## Figures and Tables

**Figure 1 life-15-00945-f001:**
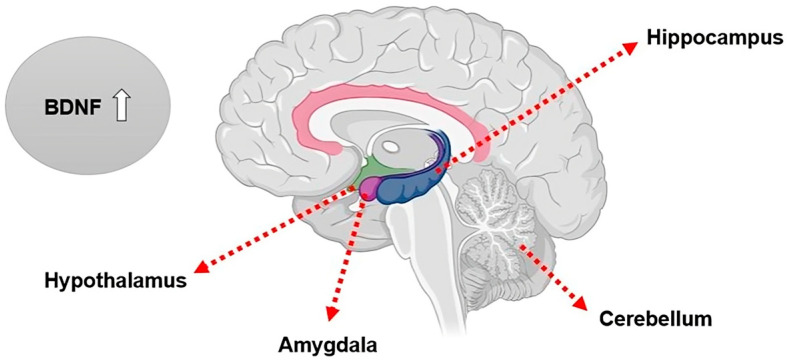
Brain regions with the highest expression levels of BDNF: the hippocampus, cerebellum, amygdala, and hypothalamus.

**Figure 2 life-15-00945-f002:**
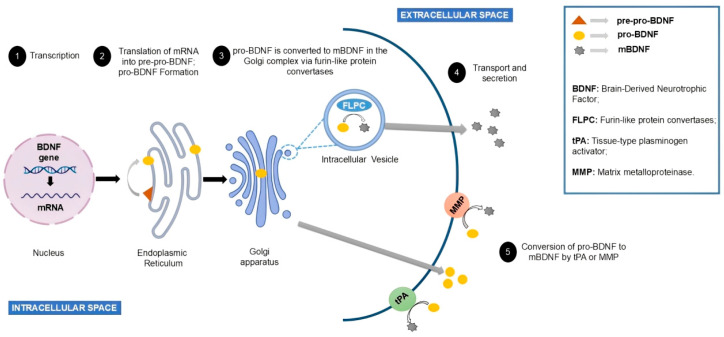
The mechanism of BDNF synthesis begins with the transcription of its gene in the nucleus, producing the corresponding mRNA, which is translated in the endoplasmic reticulum to generate pre-pro-BDNF, which will be cleaved to form pro-BDNF. After transport to the Golgi complex, pro-BDNF will undergo conversion to mBDNF by the action of furin-like protein convertases. Alternatively, pro-BDNF can be processed by the action of tissue plasminogen activator (tPA) or matrix metalloproteinases (MMPs) to generate extracellular mBDNF.

**Table 1 life-15-00945-t001:** Search strategy: descriptors and Boolean operators used for selecting eligible studies.

Search strategy
High-fat diet OR Hyperlipidic diet AND Exercise OR Physical activity AND BDNF OR Brain-derived neurotrophic factor AND Hippocampus

Source: authors [2025].

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
