# Peer review of "The Effects of the Association Between a High-Fat Diet and Physical Exercise on BDNF Expression in the Hippocampus: A Comprehensive Review"

_life, 2025, doi:10.3390/life15060945_

Round 1

Reviewer 1 Report

Comments and Suggestions for Authors

Major

1. Lines 60-78 (Methods section) The search strategy is too brief. Please specify the number of results retrieved from each database and the exact search dates. "No restrictions were applied regarding publication date" (lines 73-74) - This may introduce bias. Setting at least a starting year would be more appropriate for a focused review.

2. Lines 248-251 (HFD fat content) "fat content in experimental diets generally ranges from 45% to 60%" - This range lacks sufficient supporting references. More specific citations are needed to substantiate this claim.

3. Lines 347-350 (Molteni study results) "up to a 135% increase in BDNF levels... compared to a 76% reduction" - While these specific values are valuable, statistical significance (p-values) and standard deviations are missing, which limits the interpretation of these findings.

4. Lines 379-383 (Conclusions) The conclusions are too general. Specific exercise prescriptions (intensity, frequency, duration) based on the reviewed evidence would strengthen the clinical relevance of this review.

Minor

5. Lines 32-33 "more than 30% of total energy intake" - WHO guidelines typically use 35% as the threshold. Please clarify this discrepancy or provide the rationale for using 30%.

6. Lines 125-128 (BDNF definition) Basic information about BDNF's molecular weight and structural characteristics is missing, which would help readers unfamiliar with this neurotrophin.

7. Lines 263-268 (HFD mechanisms) The mechanisms by which HFDs reduce BDNF could be explained more clearly, particularly the relationship between oxidative stress, inflammation, and insulin resistance.

Reviewer 2 Report

Comments and Suggestions for Authors

Overall, the reading of the manuscript is enjoyable, although at times it feels somewhat repetitive. In my opinion, the paper  reports the data from the studies included in the review without offering sufficient critical analysis or commentary to enrich and deepen the discussion. It would be better to connect the various points discussed and to include personal reflections that would help make the text more original and comprehensive.

My comments are reported below.

In details:

Abstract, introduction and material and methods  are well done.

In result section, I would suggest to improve Figure 1 in 3.1, adding all informations reported in the text from line 92 to line 101.

Improve also from line 117 to line 121. I would suggest to better explain the method design in reference 12.

Also Figure 3 in 3.2 section can be improved.

Have authors evaluated gender influence on BDNF serum levels in the relationship between physical excercise and cognitive benefits?

Research indicates that males and females may exhibit different BDNF responses to physical activity. Given that,  have authors examined the influence of gender on BDNF serum levels in the context of the relationship between physical exercise and cognitive benefits?

Conclusion section could be improved.

Author Response

Thank you very much for taking the time to review this manuscript. Your comments and suggestions were very helpful in further improving the quality of the review, which often go unnoticed by authors due to constant and repetitive reading. Below you will find the detailed responses and corresponding revisions/corrections highlighted/following changes in the resubmitted files.

Comment 1: In the results section, I suggest improving Figure 1 in 3.1 by adding all the information reported in the text from line 92 to line 101.

Response 1 :

Explanation: Thank you for the suggestion to improve the image, however, it was not possible to make such a change due to the lack of copyright. Since this image of the hippocampus is an adaptation of a book, the editor advised us to remove it from the manuscript. We appreciate the suggestion, which would indeed be very great for improving understanding, but we were unable to carry it out.

Comment 2: Also improve from line 117 to line 121. I would suggest explaining the method design better in reference 12.

Response 2:
In a study involving 40 male C57BL/6 mice, 4 weeks of age, the animals were randomly divided into two groups: normal diet (CON, n = 10) and high-fat diet (HFD, n = 30). The HFD diet contained 60% fat. The animals were induced to obesity for a period of 8 weeks and subsequently randomly redistributed into three groups: HFD (n = 10), HFD with low-intensity training (HFDLT, n = 10) and HFD with high-intensity training (HFDHT, n = 10). The mice in the groups subjected to physical training performed treadmill exercises 5 times a week for 8 weeks, maintaining regular consumption of their respective diets. After the training period, an assessment of cognitive function was performed using the Y-maze test, which aims to analyze the animal's spatial memory based on the supervised exploration of the three arms of the maze. The results of the study demonstrated that the physical exercise intervention significantly improved the cognitive function of animals that consumed a high-fat diet. The number of entries and alternations was significantly higher in the physical training groups compared to the sedentary HFD group (p < 0.05) [12].

In a similar study conducted by Kim et al., 40 male C57BL/6 mice, 4 weeks of age, were also used and randomly divided into four groups: control group (CON), control group with exercise (CON + EX), high-fat diet group (HFD) and high-fat diet and exercise group (HFD + EX). Subsequently, the Y-maze test was performed, and it was observed that the
percentage of spontaneous alternation of the animals was lower in the group submitted to the high-fat diet compared to the control group [29].

These findings demonstrate that the hippocampus may be impacted by the consumption of high-fat diets, significantly compromising the animals' spatial memory. In the Y test, it was possible to naturally analyze the rodent's exploratory capacity, observing the order of the input sequence, identifying the number of hits or misses during the test. With the consumption of HFD, BDNF levels tend to decrease, which results in impairment of cognitive capacity related to learning and memory. On the other hand, physical training has proven to be increasingly efficient in minimizing the damage caused by this type of diet.

Explanation: Thank you for your comment, it was extremely important to enrich the data presented in the study. The suggested change was accepted by the authors and modified in the manuscript and can be found at the end of page 3 (lines 123-150). In addition to explaining the method used by reference 12, a similar study was also added to complement this thought, in addition to a contextualization at the end of the paragraph.

Comment 3: Figure 3 in section 3.2 could also be improved.

Response 3:

Figure 2. The mechanism of BDNF synthesis begins with the transcription of its gene in the nucleus, producing the corresponding mRNA, which is translated in the endoplasmic reticulum to generate pre-pro-BDNF, which will be cleaved to form pro-BDNF. After transport to the Golgi complex, pro-BDNF will undergo conversion to mBDNF by the action of furin-like protein convertases. Alternatively, pro-BDNF can be processed by the action of tissue plasminogen activator (tPA) or matrix metalloproteinases (MMPs) to generate extracellular mBDNF.

Explanation: Thank you for the suggestion to improve the image, which became Figure 2 with the change. In addition to adding the steps of BDNF synthesis, its caption was also improved. With your help, a layperson can understand this process more easily. These changes can be found on page 5 (lines 167-174).

Comment 4: Did the authors examine the influence of gender on serum BDNF levels in the relationship between exercise and cognitive benefits? Research suggests that men and women may have different BDNF responses to physical activity. Given this, did the authors examine the influence of gender on serum BDNF levels in the context of the relationship between exercise and cognitive benefits?

Response 4:

Explanation: Thank you for this suggestion. It is indeed a very important point in the scientific community, since men and women react differently to physical exercise, presenting possible physiological changes in the organism. However, the articles that were mentioned in the manuscript did not evaluate the influence of gender on serum BDNF levels, reporting only its benefits for cognitive processes. Thank you again for this suggestion, but unfortunately we were unable to modify it.

Comment 5: The conclusion section could be improved.

Performing physical activity on the treadmill, 5 times a week, for 8 weeks, was the most commonly used duration among the studies, with no relevant differences between low and high intensity exercises, proving that both intensities help prevent cognitive defi-cits, especially in the hippocampus.

Explanation: Thank you for pointing this out and we agree with this comment. The changes can be found on page 11 (lines 446-449). Most of the studies mentioned used a 5 day/week exercise training regimen for 8 weeks, which is believed to be a standard protocol among studies to find significant results.

3. Additional clarifications

We are available to answer any questions that may arise. Once again, we thank you for taking the time to review this manuscript. We hope that the changes are as requested and thank you for your contributions.

Round 2

Reviewer 2 Report

Comments and Suggestions for Authors

The paper has been improved and is suitable for being published.